# The Advances in Epigenetics for Cancer Radiotherapy

**DOI:** 10.3390/ijms23105654

**Published:** 2022-05-18

**Authors:** Yuexuan Wang, Yu Han, Yuzhen Jin, Qiang He, Zhicheng Wang

**Affiliations:** NHC Key Laboratory of Radiobiology, School of Public Health, Jilin University, Changchun 130021, China; wyx15960204504@163.com (Y.W.); hy15568136829@163.com (Y.H.); jinyz5079@163.com (Y.J.)

**Keywords:** radiotherapy, epigenetic modification, DNA methylation, histone modification, chromatin remodeling, RNA modification

## Abstract

Cancer is an important factor threatening human life and health; in recent years, its morbidity and mortality remain high and demosntrate an upward trend. It is of great significance to study its pathogenesis and targeted therapy. As the complex mechanisms of epigenetic modification has been increasingly discovered, they are more closely related to the occurrence and development of cancer. As a reversible response, epigenetic modification is of great significance for the improvement of classical therapeutic measures and the discovery of new therapeutic targets. It has become a research focusto explore the multi-level mechanisms of RNA, DNA, chromatin and proteins. As an important means of cancer treatment, radiotherapy has made great progress in technology, methods, means and targeted sensitization after years of rapid development, and even research on radiotherapy based on epigenetic modification is rampant. A series of epigenetic effects of radiation on DNA methylation, histone modification, chromosome remodeling, RNA modification and non-coding RNA during radiotherapy affects the therapeutic effects and prognosis. Starting from the epigenetic mechanism of tumorigenesis, this paper reviews the latest progress in the mechanism of interaction between epigenetic modification and cancer radiotherapy and briefly introduces the main types, mechanisms and applications of epigenetic modifiers used for radiotherapy sensitization in order to explore a more individual and dynamic approach of cancer treatment based on epigenetic mechanism. This study strives to make a modest contribution to the progress of human disease research.

## 1. Introduction

In recent years, cancer is eroding human life, and its occurrence and development is closely related to genetic and epigenetic changes. At the same time, the emergence of more and more cancer patients and the diversity of cancer also force people to adopt a variety of methods to treat and prevent cancer. Currently, surgery, radiotherapy and chemotherapy are the main methods for cancer treatment. Additionally, immunotherapy has gradually developed. Radiotherapy is one of the main treatments currently in use, and about 50% of cancer patients receive radiotherapy [1]. Radiotherapy involves DNA damage caused by ionizing radiation (IR), which aims to damage or kill cancer cells [2]. Radiotherapy can be used alone or in combination with other treatments. Meanwhile, radiation treatment will inevitably cause damage to normal tissue around the cancer [1]. Radiotherapy can remove cancer cells in various ways: It can either directly act on the DNA or indirectly cause DNA damage through free radicals produced by IR. In addition to genetic changes to DNA of cells, IR can also affect their epigenetics.

Epigenetics refers to a stable heritable phenotype caused by chromosomal changes without alterations to the DNA sequence [3]. Epigenetic modifications such as DNA methylation, histone modification, chromatin remodeling, RNA modification and changes in non-coding RNAs, including miRNA, govern epigenomic alterations (Figure 1) [4]. Studies have found that the occurrence of cancer is closely related to changes in epigenetics [3]; meanwhile, IR can achieve the purpose of treatment by intervening the epigenetics of some types of cancer. Therefore, the study of epigenetics is of great significance to cancer. With deepening research, it was observed that some epigenetic modifiers can be used as radiation sensitizers, such as histone deacetylase(HDAC) inhibitors, DNA methyltransferase (DNMT) inhibitors, EZH2 inhibitors BET inhibitors, etc. They can destroy DNA-damage repair and cell cycle and increase oxidative stress to enhance the anti-tumor activity of radiotherapy [5]. The drugs can be used as an important partner for cancer radiotherapy. In this review, we will summarize the relationship between epigenetics and cancer occurrence, and we explore some progress in radiotherapy based on epigenetics.

## 2. Epigenetics

Epigenetics was first proposed by Conrad H. Waddington, a Scottish embryologist and geneticist. With the development of modern science, epigenetics has developed from an obscure beginning into a widely recognized branch of biology. The concept of epigenetics has changed by time [6]. There is also a greater understanding of the molecular mechanisms underlying epigenetic regulation.

DNA methylation was discovered by Hotchkiss (1948) in calf thymocytes. It refers to the chemical modification process in which specific bases on DNA sequence obtain a methyl group by covalent bonding with S-adenosine methionine (SAM) as methyl donor under the catalytic action of DNMT. DNA methylation is a kind of heritable symmetric epigenetic mark, which almost only exists on the cytosine residue carbon 5 in high eukaryotes, and the main target of methylation is CpG dinucleotide [7]. Histone modification was first proposed in 1950. The main covalent histone modifications are acetylation, methylation, phosphorylation, ADP-ribosylation, ubiquitination, SUMOylation, citrullination, glycosylation, hydroxylation and isomerization. Acetylation, methylation and phosphorylation are best studied in the context of gene expression regulation, chromatin structure establishment, replication and DNA repair [7]. Chromatin needs to undergo a series of changes during gene expression. For example, in eukaryotes, nucleosomes are the basic unit of chromatin. Because genomic DNA is tightly wrapped around the histone octamer in the nucleosome, its function is severely limited in chromatin. In order to overcome the nucleosome barrier, nucleosome structure must change dynamically during genomic DNA function [8], such changes in chromatin structure that occur during gene expression regulation are called chromatin remodeling. RNA modification has always been considered as a relatively static trimmer of RNA structure and function. However, studies have shown that RNA modification is reversible and dynamically regulated, and the activity of RNA modification can be regulated by many factors. Approximately half of mutations in RNA modifying enzymes are known to be associated with human diseases, including cancer, cardiovascular disease, inherited birth defects, metabolic diseases, neurological diseases and mitochondria-related defects [9]. The central tenet of molecular biology states that RNA’s function revolves around protein translation. Until the last decade, most studies focused on characterizing RNA as a mediator of protein translation, emphasizing the functions of mRNA, tRNA and rRNA. However, these processes account for less than 2% of the genome and are not sufficient to explain 98% of the functions of transcribed RNAs. Recent discoveries have revealed thousands of unique non-coding RNAs (ncRNAs) and have changed the view of them from “junk” transcripts to “unexplained” and are potentially extremely important. In major cancers, ncRNAs have been identified as carcinogenic drivers and tumor suppressors, suggesting a complex regulatory network between these ncRNAs [10]. Thus far, studies have shown that epigenetic changes are not only related to the occurrence of cancer but they can also be affected by radiotherapy on the epigenetic changes of cancer cells.

## 3. Epigenetics in Cancer Occurrence and Cancer Radiotherapy

### 3.1. DNA Methylation

#### 3.1.1. DNA Methylation and Carcinogenesis

DNA methylation is the earliest and most widely studied aspect of epigenetics. Increasing evidence shows that DNA methylation is closely related to the occurrence and development of cancers [11]. CpG islands are located in the promoter region and transcriptional initiation site of the gene, which are prone to DNA methylation and change the expression of oncogenes and tumor suppressor genes [12]. In the development of cancer, the methylation of specific gene promoter is considered to be a predictor of radiosensitivity and a prognostic factor. The typing of molecular subtypes based on DNA methylation is also closely related to the clinical characteristics of some cancers [13]. The hypermethylation of CGI in the promoter region leads to the silencing of tumor suppressor genes, while the overall hypomethylation of repetitive elements leads to the reactivation of retrotransposons, both of which affect the stability of the genome and become risk factors for cancer (Figure 2a) [14].

Promoter hypomethylation is the key mechanism of the upregulation of SLCO4A1-AS1 in colorectal cancer (CRC). SLCO4A1-AS1 promotes the occurrence of CRC by strengthening the binding of Hsp90 and Cdk2 (Figure 2a) [15]. UNC5 receptor inhibition mediated by the hypermethylation of the promoter plays an important role in the occurrence and progression of CRC (Figure 2a) [16]. ZBTB28 is highly expressed in normal renal tissue, but it is significantly downregulated in renal cell carcinoma (RCC) cell lines because its promoter is often methylated. As a tumor suppressor gene, ZBTB28 can inhibit cell proliferation and metastasis and promote apoptosis. Its silencing induces the development of RCC (Figure 2a) [17]. The increase in UHRF1 in cervical cancer (CC) tissue promotes the hypermethylation of the TXNIP promoter to downregulate the expression of TXNIP, thus promoting carcinogenesis (Figure 2a) [18]. SULT2B1 can inhibit the proliferation of esophageal squamous cell carcinoma (ESCC) cells, and the hypermethylation of its promoter can promote the progression of esophageal tumor by downregulating PER1 (Figure 2a) [12].

We know that some cancers are thought to be associated with specific viruses, and several recent studies have shown that epigenetic mechanisms may be involved in the process of virus inducing cancer and the progression of cancer. LMP1 encoded by EBV can inhibit the expression of RASSF10 and induce DNA methylation of RASSF10 by recruiting DNMT1 to promote tumorigenesis (Figure 2a) [19]. In another study, the difference in hTERT methylation patterns in CC specimens was related to the type of HPV [20].

#### 3.1.2. Radiotherapy and DNA Methylation

Methylation or demethylation of DNA and changes of DNMT may be induced during radiotherapy [21]. The characteristics of its changes in different cancer are also important factors to comprehensively evaluate whether the treatment regimen can improve the quality of life of patients. With the progress of treatment technology and the increase in long-term survival rates, the risk of secondary diseases induced by radiotherapy has been paid more and more attention [22]. In the study of rat breast cancer, it was found that the radiation-induced polycomb repressive complex 2 (PRC2)-mediated DNA hypermethylation of transcription factors may lead to cancer cell dedifferentiation and participate in radiation-induced breast cancer (Figure 2b) [22]. 

Previous studies have shown that IR can induce cognitive impairment [23]. Recent studies have found that the expression of DNMT1 and other methylases related to consciousness decreased after craniocerebral irradiation, suggesting that the occurrence of disturbance of consciousness after radiotherapy may be related to the epigenetic mechanism (Figure 2b) [24,25]. Another study found that in childhood cancer survivors (CCS, >16 years old) after receiving total body irradiation (TBI) and hematopoietic stem cell transplant (HSCT), CD4+ and CD8+T cells tended to show lower overall DNA methylation levels, while monocytes tended to show higher methylation levels, suggesting that TBI/HSCT may be related to long-term immune disturbances (Figure 2b) [26].

In addition to radiation-induced complications, the efficacy of radiotherapy is also a point of concern, and it is closely related to the radiosensitivity of tumor cells. Different epigenetic characteristics may exist before irradiation, but during treatment, specific DNA methylation changes may also be involved in the mechanism of regulating cell survival, thus forming a target for radiosensitization [27]. Therefore, in the individualized treatment of cancer, the detection of epigenetic status before and during radiotherapy is of great significance. A recent study found that DNMT3B silencing restores the function of p53 and p21 through DNA demethylation, which induces cell cycle arrest and apoptosis. IR can induce the increase in DNMT3B and the methylation of p53 and p21 to promote the radiation resistance of nasopharyngeal carcinoma (Figure 2b) [28]. The average methylation percentage of DAPK1 and BRCA1 genes decreased and the transcriptional levels of BRCA1 and DAPK1 genes increased significantly in the samples of CC patients before and after radiotherapy and chemotherapy with 10 Gy. BRCA1 is involved in the repair of damaged DNA, while DAPK1 is a pro-apoptotic gene and may inhibit metastasis (Figure 2b) [29]. 

Studies have shown that radiation may induce the increase or decrease in miRNA expression [30], which may also be related to the methylation status of its upstream promoter. In a study, TET2 gene promotes radiation-induced demethylation of miR-378a promoter through interactions with ATF2, while miR-378a-3p can reduce the cytotoxicity of NK cells by inhibiting the expression of granzyme-B (Figure 2b) [31]. Interestingly, another kind of miRNA, miR-10b, was found to be overexpressed in esophageal squamous cell carcinoma and CRC [32,33], and it has a role in promoting cancer invasion and progression. However, in some cancers, such as gastric cancer, it may be found as a new tumor suppressor and partially silenced by hypermethylation of DNA in gastric cancer [34]. In a study of thyroid cancer, miR-10b-5p reduced the growth rate and viability of irradiated 8505c cells (derived from human thyroid cancer) [35]. More mechanisms of the interaction between cancer radiotherapy and miRNA will be explained in more detail in the RNA section below.

### 3.2. Histone Modification

#### 3.2.1. Histone Modification and Carcinogenesis

Histone modification is another very important part of epigenetics, including histone methylation, acetylation, phosphorylation and ubiquitin. Histone modification regulates gene expression, which is closely related to transcription and DNA repair [36,37,38]. In these processes, histone modification also interacts with DNA methylation [39]. 

The change of histone acetylation level is closely related to the occurrence of some tumors, such as deacetylation may promote tumorigenesis by down-regulating the expression of tumor suppressor genes. A recent study reported that the decrease in H3K9ac levels regulates the expression of related genes such as proliferation and migration of pancreatic cancer cells through the Ras-ERK1/2 pathway [40]. In sporadic parathyroid adenomas, promoter methylation and H3K9 deacetylation together lead to the silencing of tumor suppressor gene PAX1 [41]. Another study examined the difference in H3K27ac levels between papillary thyroid carcinoma (PTC) and benign thyroid nodules (BTN) and found that the changes may be related to the occurrence and prognosis of papillary thyroid cancer [42]. 

The role of histone methylation and its related enzymes in tumorigenesis has also been widely reported. A recent study shows that H3K36me2 plays a key role in tumorigenesis [43]. However, some histone methyltransferases, such as SETD8, have been found to be significantly overexpressed in RCC and can be positively correlated with tumor grade and stage as a prognostic factor [44]. Another histone methyltransferase NSD2-mediated H3K36me2 and carcinogenic Ras signal pathway synergistically drive the development of lung adenocarcinoma (LUAD) and play an important role in transcriptional activation and expression of multiple oncogenes in CRC [45,46]. The complete loss of H3K27me3 indicates an increased risk of meningioma recurrence [47].

#### 3.2.2. The role of Histone Modification in Radiotherapy

Bulleted lists look like this: It has been widely reported that radiation has a series of effects on histone modification in tumor cells and normal cells [48]. H3K27me3 is related to chromatin condensation, which affects DNA double-strand breaks (DSB) repair. In one study, it was shown that radiation induced H3K27me3 loss, while GSKJ4 (H3K27 demethylase inhibitor) inhibited radiation-induced DSB repair and enhanced radiosensitivity of tumor cells, while histone demethylase UTX reduced radiosensitivity [49]. 

In another study of diffuse intrinsic pontine glioma (DIPG), the abundance of H3K9me3 increased in all four cell lines after radiotherapy. Radiotherapy combined with histone methyltransferase G9a inhibitor was designed to reduce H3K9me3 levels and DSB repair. Radiotherapy can induce some histone-modified cell-specific and peptide-specific changes, which should be paid attention to when using modifiers [50]. It has been suggested that radiation kills a large number of cancer cells and induces radiation resistance and more aggressive epigenetic phenotypes. In this study, there are residual cells in human hepatocellular carcinoma xenografts after radiotherapy, and the up-regulation of CXCL12 mediated by histone modification in its promoter may be an important reason for the occurrence of resistant phenotype in treatment. Interestingly, in co-cultured Huh7 cells, radiation-induced CXCL12 mRNA seems to be larger than that of single cultured cells, indicating that co-culture of tumor and normal cells is beneficial to tumor survival. At the same time, it was found that IOX1 (histone demethylase) may inhibit radiation-induced CXCL12 [51]. 

Some studies have suggested that radiation-induced autophagy is also related to histone methylation, and H4K20me3 is the key to induce autophagy after irradiation. Radiation-induced autophagy is a protective mechanism of non-small cell lung cancer (NSCLC) cells. The inhibition of autophagy-associated histone modification will increase cell death after radiation. DZNep (3-deazaneplanocin A), a broad-spectrum methyltransferase inhibitor, can inhibit H4K20me3, and its treatment significantly increased radiosensitivity [52]. Irradiation triggered methylation of histone H3 on the promoter of aldehyde dehydrogenase 1A1 (ALDH1A1), a CSC marker in prostate cancer cell line, thus stimulating its gene transcription. DZNep can also target histone 3 methylation, resulting in the downregulation of ALDH1A1 expression and radiosensitivity of tumor cells, and the anti-radiation effect is more obvious [53]. 

Zebrafish has 70% genetic similarity with humans and has become a widely used organism in radiation research [54]. When zebrafish embryos were exposed to 10.9 mGy/h gamma rays for 3 h, the levels of H3K4me3, H3K27me3 and H3K9me3 in all studied genes were higher than those in the control group, indicating that these histone PTMs may be potential biomarkers of IR. Another study of Atlantic salmon embryos in similar exposed environments showed that the enrichment of H3K4me3 at the same site was conserved between two species far apart in evolution [55]. It also provides valuable clues and new ideas for the study of human epigenetics.

Histone acetylation activates gene transcription through the acetylation of lysine amino acid residues on the histone tail. At 1 and 30 days after cranial irradiation, the total level of acetylation of histone H3 decreased significantly [25]. In another experiment, there was quite long-lasting and extensive deacetylation of lymphoblasts after radiation. However, there were differences among individuals, and radiation-sensitive cell lines showed more obvious and lasting H4K16 deacetylation [56]. Another recent study reported that radiation-resistant populations showed overall histone deacetylation and changes in HDACs and histone acetyltransferases(HATs) activity. However, HDAC activity also exists heterogeneity among different individuals. Tumor HDAC activity should be evaluated before radiotherapy. Patients with high activity are suitable for radiosensitization with histone deacetylase inhibitors (HDACi) [57]. 

### 3.3. RNA Modification and Non-Coding RNAs

#### 3.3.1. RNA Modification and Carcinogenesis

Post-transcriptional modifications of RNA molecules are widely available. Currently, there are more than one hundred known RNA modifications, such as N6-methyladenine (m^6^A), 5-methylcytosine, N1-methyladenosine and M7G, among which methylation modification is the most common. m^6^A is one of the most common post-transcriptional modifications of RNA in eukaryotes. Moreover, it can perform important functions that affect normal living activities and disease. Most studies have shown that m^6^A can influence the complexity of cancer progression by modulating cancer-related biological functions [58]. m^6^A is present in mRNA, lincRNAs, pri-miRNA and rRNA [59], and it is involved in various aspects of RNA metabolism including mRNA splicing, 3′-terminal processing, translation regulation, mRNA decay and non-coding RNA processing. m^6^A is mainly regulated by three regulators, namely methyl transferase (METTL3, METTL14 and METTL16, etc.), demethylation transferase (FTO and ALKBH5) and methyl recognition protein (YTHDF1, YTHDF2, YTHDF3, YTHDC1 and YTHDC2, etc.). As “writer”, “eraser” and “reader”, respectively, they are proteins that can add, remove or recognize m^6^A sites and alter important biological processes accordingly [60]; when they are disordered, they are closely associated with cancer initiation and progression.

The high expression of methyltransferase-like 3 (METTL3) can promote tumor angiogenesis in gastric cancer tissues [61]. In contrast, METTL14 inhibits the occurrence and progression of gastric cancer by regulating the pathway [62]. In acute myeloid leukemia, m^6^A demethylases, FTO plays a key role in the development of leukemia. Some oncogenic proteins, such as MLL fusion protein, can up-regulate FTO, thus enhancing the activity of AML cells, promoting proliferation and inhibiting apoptosis [63]. In pancreatic cancer, METTL3 protein and mRNA levels are significantly increased, which promote cell proliferation, invasion and migration [64]. In CC, METTL3 is significantly upregulated in CC, which is closely associated with lymph node metastasis and poor prognosis in patients. METTL3 enhances the stability of HK2 (hexalokinase 2) through YTHDF1-mediated m^6^A [65], and FTO is often overexpressed in human CC tissues and is highly associated with promoting the progression of CC [66]. In addition, the occurrence of other cancers is also closely related to the dysregulation of regulatory proteins, such as endometrial cancer [67], glioblastoma, etc., [60].

#### 3.3.2. Effects of Radiotherapy on RNA Modification

RNA modification of tumor epigenetic also has an impact on radiotherapy, which is also an important part of our research, in order to obtain new ideas and directions in radiotherapy. In nasopharyngeal carcinoma (NPC), YTHDC2 is highly expressed in radiation-resistant NPC cells. The knockout of YTHDC2 can improve therapeutic effects of radiotherapy in vitro and in vivo. While the overexpression of YTHDC2 in radiation-sensitive NPC cells is the opposite [68]. Experiments showed that the expression of FTO was increased in cervical squamous cells. After radiotherapy, it is found that the overexpression of FTO would increase the survival rate of tumor cells and have a certain radioresistance [69]. In pancreatic cancer, METTL3 knockdown cells are highly sensitive to radiation, so it can be speculated that METTL3 levels will affect radiotherapy [70]. In glioblastoma, METTL3 expression increased, and for METTL3-silenced glioma stem cells (GSCS), their radiosensitivity was enhanced and DNA repair was reduced. METTL3 can be used as a potential molecular target for GBM therapy and METTL3 alters the DNA repair efficiency and radiation sensitivity partially via SOX2 in GSCs [71]. In hypopharyngeal squamous cell carcinoma (HPSCC), METTL3 mediates m^6^A methylation and stabilized the expression of circCUX1, a specific circRNA, and knocking down circCUX1 promotes the sensitivity of hypopharyngeal cancer cells to radiotherapy. In addition, circCUX1 binds to caspase1 and inhibits its expression, resulting in reduced release of inflammatory cytokines and, thus, tolerance to radiotherapy [72].

#### 3.3.3. Non-Coding RNAs Are Closely Related to the Occurrence and Development of Cancer

Non-coding RNA (ncRNA) is an RNA that does not encode a protein, which is not translated into a protein, and can perform their respective biological functions at the level of RNA. It can be divided into three categories: less than 50 nucleotides (nt), including microRNA, siRNA and piRNA; 50 nt to 500 nt, including rRNA, tRNA, snRNA, snoRNA, SLRNA and SRPRNA; more than 500 nt, including long mRNA-like non-coding RNA, etc. [73,74,75]. Micro-RNAs (miRNAs) is a short RNA molecule of 19–25 nt in length. They can restrain translation and trigger the degradation of the target RNA, which plays an important role in regulating gene expression [76].

MiRNAs are able to target hundreds of transcripts as the regulators. They are very powerful so their abnormal expression can disturb massive cellular signaling pathways and affect cancer initiation and progression [77]. MiRNAs is a potential regulator of the oncogenic potential of urinary bladder cancer (BC) cells. MiR-23a-3p is involved in the cancerous network of BC through up-regulation [78,79]. The up-regulation of miR-21 and miR-9 promote cancerous progression of BC and cause the poor patient prognosis (Table 1) [78,80]. The down-regulation of miR-495 enhances the value-added and invasiveness of BC [81]. In ovarian cancer (OC), miR-135a-3p, miR-200c, miR-216a and miR-340 can regulate the invasiveness of OC cells by regulating the epithelial-mesenchymal transition (EMT) program [82,83,84,85]. In addition, miRNAs are closely associated with the occurrence and development of many malignant tumors, such as lung cancer, pancreatic cancer and breast cancer [86,87,88,89,90,91].

Studies have shown that m^6^A regulatory proteins can regulate m^6^A modification in ncRNA to produce various biological functions; for example, METTL3 can promote miR-221/222 processing and BC cell proliferation; in turn, ncRNA can also regulate m^6^A methylation of mRNA in cancer. For example, miRNA regulates m^6^A formation on mRNA and promotes cell reprogramming into pluripotent stem cells [107].

#### 3.3.4. Influences of Radiotherapy on Non-Coding RNA

Bulleted lists look like this: Currently, radiotherapy has become the main means of cancer treatment, and miRNAs are also widely used. In the radiotherapy of CRC, miRNAs are involved in regulating the radiosensitivity of CRC cells, where miR-195 suppresses the expression of CARM1 and enhances the radiosensitivity of CRC cells; miR-124 can improve the radiosensitivity of CRC cells by blocking the expression of PRRX1 [108,109,110]. Saeid Afshard et al. detected the up-regulation of miR-185 could down-regulate the expression of IGF-1R and IGF2 and increased the sensitivity of CRC cells to IR by transferring miR-185 cells [111]. IGF-1R protein was amplified in CRC, and Pouria Samadi et al. found that up-regulation of let-7 of the miRNAs’ family could inhibit the expression of IGF-1R and increase radiosensitivity (Table 1) [100].

However, not all miRNAs could increase the IR sensitivity of CRC cells. The overexpression of miR-106b reduces the expression of PTEN and p21 and induces resistance to IR in poorly differentiated cells in vitro and in vivo. Moreover, radiation observations in lung cancer show that the low expression of miRNA-21 inhibits the proliferation of A549 cells and sensitizes the cells to IR. The high-level expression of miRNA-214 is related with the resistance of NSCLC cell lines to radiation and protects the cells from radiation-induced apoptosis [112,113]. In breast cancer, miR-122 can reduce the survival of cancer cells and promote their sensitivity to IR, whereas the overexpression of miR-122 is associated with radiation resistance [114]. LncRNA HOTAIRM1 (a type of long non-coding RNA) plays a role in glioblastoma proliferation and invasion, and its downregulation can regulate glioblastoma radiosensitivity in vitro and in vivo [115].

Thus, it shows that different expression forms of miRNAs in cancer may cause different effects, some leading to increased sensitivity to radiation therapy and others leading to radiation resistance, which makes treatment more difficult.

### 3.4. Chromosomal Remodeling

#### 3.4.1. Chromosome Remodeling and Cancer

Nucleosomes, the basic structural unit that constitutes the chromatin, consist of approximately 146 base pairs of DNA surrounding a histone octamer core that contains two molecules of the core histones H2A, H2B, H3 and H4. The linear arrays of nucleosomes of chromatin primary structures fold and condense to varying degrees in the nucleus and chromosomes to form “advanced structures” [116,117]. Chromatin remodeling refers to the packaging state of chromatin, histones and corresponding DNA molecules in nucleosomes in the process of replication and recombination of gene expression. Chromatin remodeling proteins aim to alter nucleosome structure to expose genes that are hidden in transcriptional machinery. Nucleosome reorganization can be performed by two mechanisms: by ATP-dependent chromatin remodeling complexes and modification of core histones by histone acetyltransferases, deacetylases, methyltransferases and kinases [118,119]. Enzymes associated with chromatin remodeling play important roles in the repair of DNA damage, gene expression and transcription. Thus, damage to chromatin remodeling is closely related to the development of cancer [38,120,121].

Chromatin remodeling factors have four major families, including SWI/SNF, ISWI, Mi-2 and IN080 families. Alterations of their many subunits (e.g., inactivating mutations, homozygous deletions, silencing and overexpression) are associated with cancer occurrence and development [38,122]. Among them, SWI/SNF is a widely studied remodeling factor, and the inactivating mutation of its core subunit SNF5 is very common in malignant rhabdoid cancer, and attenuated SNF5 expression also promotes BC progression through the activation of the STAT3 gene [123,124]. There are also many malignancies tumorigenesis associated with SWI/SNF, such as the association between Ini1 changes and clear cell renal cell carcinoma, atypical central system teratomas [125,126], mutations in the BAF250A gene are associated with OC, endometrial cancer, liver cancer, etc. [127,128,129,130]. The overexpression of MTA1 in the Mi-2 family compared with NPC, carcinoma of uterine cervix, pancreas cancer and so on [131,132,133,134].

#### 3.4.2. Changes in Chromosome Remodeling during Radiotherapy

Chromatin remodeling is very important in cancer. Radiotherapy treats cancer by using IR to induce DSB in cancer. Targeting BRG1 chromatin remodeling enzymes can improve the radiosensitivity of many cancers (such as CRC). BRG1-BRD makes cancer cells more sensitive to IR, which pass BRG1 explicit negative activity, thereby destroying the γ-H2AX and 53BP1 pathway. This can cause the low efficiency of DNA repair, G2-M checkpoint defects and enhance apoptosis [135]. Chromatin remodeling protein MORC2 can be acetylated by NAT10 to regulate DNA damage, G2 checkpoint capture through acetylation and improve radiosensitivity in breast cancer [136]. More experiments and methods targeting chromatin remodeling in radiotherapy are still being updated and studied, and they will be more widely used in clinical practice.

## 4. Application of Epigenetic Modifiers in the Sensitization of Radiotherapy

Certain drugs that can be selected to make cancer more sensitive to IR are described as radiation modifiers because of their ability to alter the cancer response to irradiation [137]. Radiosensitizers are used to describe drugs that selectively enhance radiation killing to cancer cells and do not exhibit single-drug toxicity to cancer or normal tissues [138]. In the treatment process of cancer inhibition by radiotherapy, the application of epigenetics is very important, providing targets and ideas, and it can function as a sensitizer for the combination therapy of cancer.

Currently, HDACi can play the role in radiotherapy sensitization by inducing apoptosis, inhibiting DSB repair, blocking cell cycle and other multiple ways, radiotherapy may also affect the number and activity of HDAC to a certain extent (Figure 3) [139]. HDACi has entered the clinical trials of radiotherapy for various tumors (Figure 3) [140,141]. SAHA (vorinostat), an HDACi, could significantly increase radiosensitivity compared to treatment alone and could significantly delay lung metastasis by inhibiting MMP-9 activity in vivo and in vitro [142,143]. Entinostat is also an HDACi; McLaughlin et al. found that, in a model of NSCLC, entinostat could sensitize a proportion of NSCLC cells to IR by effectively down-regulating FLIP expression and the capacity to boost caspase-8 activation [144]. TSA is also used as an HDACi; the results showed that TSA pretreatment could enhance DNA damage induced by IR and make esophageal cancer cells more sensitive [145]. Cisplatin therapy is a common clinical treatment for tumor chemotherapy. The combination of SAHA and cisplatin can re-sensitized cisplatin resistant bladder cancer cells. Cisplatin resistant human bladder cancer cell lines can be re-sensitized through cell cycle arrest and the induction of Caspase-dependent apoptosis [142,146]. In addition, Seiichiro Komatsu et al. demonstrated that SAHA combined with bortezomib and clarithromycin showed therapeutic support for breast cancer cell lines [147].

Other sensitizer effects are DNMT inhibitors, which can increase chromatin accessibility and radiation sensitivity through chromatin breakdown or complete combination therapies with radiation by promoting DNA damage and inducing apoptosis (Figure 3) [38,148]. In clinical trials on OC, the use of decitabine can show the biological activity of low DNA methylation and improve the sensitivity of cancer cells during radiotherapy, largely resolving the late insensitivity of OC therapy (Table 2) [149]. DNMT inhibitor has also been shown to re-sensitized cisplatin in platinum-resistant ovarian cancer patients [148]. In a related phase II clinical trial, it was shown that low doses of Decitabine altered the DNA methylation of genes and cancer pathways. Sensitivity to carboplatin was restored in patients with ovarian cancer who had received extensive pretreatment [150,151].

Sensitizers related to RNA modification are also widely studied, such as the target of m^6^A methylation of RNA; the METTL3-METTL14 heterodimer is involved in the biological behavior of malignant bone marrow cells and glioblastoma, breast cancer, hepatocellular carcinoma, leukemia, etc. The expression levels of demethylase FTO and methylase (METTL3 and WTAP) are positively correlated with cancer cell resistance to chemotherapy and radiotherapy, and the downregulation of these genes can increase the radiosensitivity of glioma, pancreatic cancer and colon cancer [159].

Cisplatin-based therapy is the recommended therapy for locally advanced CC, Mengdong Ni et al. by conducting in vivo and in vitro experiments confirmed that bromodomain protein 4 (BRD4) can combine with the promoter area of RAD51AP1. It can accelerate RAD51AP1 transcription, and BRD4 inhibitors can inhibit RAD51AP1 transcription by inhibiting BRD4 activity, leading to significant radiosensitization and enhancement of CC cells (Table 2) [157]. In addition to the above pathways, BRD4 can also regulate the expression of PD-L1 by binding to CD274 acetylated histones, and indirectly play a role in regulating tumor growth [160]. In the treatment of NSCLC, radiotherapy-induced upregulation of PD-L1 leads to treatment resistance and treatment failure (Figure 3). Both JQ1 and ARV-771 belong to BRD4 inhibitors, which may enhance chemotherapy sensitivity and anti-tumor immunity of chemoradiotherapy by reducing treatment-induced PD-L1 expression in NSCLC (Table 2) (Figure 3) [156].

There are also many epigenetically modified targets associated with radiotherapy resistance as the potential radiosensitization strategy. For example, histone lysine demethylase 4C (KDM4C) can encode related histone demethylases, and its de-ubiquitination improves radiotherapy resistance in lung cancer cells. Therefore, there is likely to be a potential association between targeted KDM4C and radiosensitization, which is still under investigation [161].

Thus far, there are still many epigenetically targeted drugs that are used as sensitizers during the radiotherapy, while some are radiotherapy resistant. There are also many drugs based on epigenetic basis that are still being discovered and tested, providing more methods and feasibility to improve the therapeutic effect of cancer and avoiding its drug resistance and the insensitivity of chemotherapy.

## 5. Summary and Prospect

Currently, many research studies on the relationship between epigenetics and cancer have been conducted, and a substantial amount of progress has been made. Our review focuses on the latest advances in the epigenetic mechanism of carcinogenesis and the interaction between epigenetic modification and cancer radiotherapy. Moreover, the latest findings of some modifiers for radiosensitization based on epigenetic targets are listed. Epigenetics has many individual and tissue differences, which limits the scope of application of some conclusions. However, we believe that with the progress of science and technology and the emergence of more and more extensive studies, treatments based on the combination of epigenetics and radiotherapy will play a greater role. Based on this paper, we draw the following three conclusions.

**(1) The study of epigenetics contributes to the discovery of new pathways for tumorigenesis.** With the more in-depth and extensive study of cancer, we found that both epigenetics and genetic changes are important factors in the occurrence and development of cancer. More and more studies have found that the mechanism of cancer induced by exposure to drugs, viruses, organic chemicals and other risk factors may be related to various epigenetic pathways. This may provide a new target for us to prevent or reduce the harm caused by various risk factors.

**(2) The combination of reversible regulation and radiotherapy may lead to a better prognosis.** As a reversible regulation, epigenetic regulation also brings more possibilities for the treatment of cancers. According to the current research on cancer epigenetics, it has been found that many targets can be combined with radiotherapy to change cell radiosensitivity and disease prognosis. The epigenetic code is complex and numerous, but with the emergence of more studies on its molecular mechanisms, this complex information network is bound to introduce greater surprises to cancer treatment.

**(3) Individualized and dynamic tumor therapy based on epigenetic mechanism has a bright future.** For the research of cancer radiotherapy, individualized specific therapy has garnering increasing attention. The radiation resistance of cancer cells is dynamic in nature [53]. Epigenetic differences may exist before radiotherapy, but both radiation resistant phenotypes and radiation sensitive phenotypes may change again during radiotherapy. As a result, the prognosis of the tumor changes continuously with various epigenetic changes induced by repeated treatment and radiation. This makes the treatment of cancers more complicated, but it also provides more new treatment ideas.

## Figures and Tables

**Figure 1 ijms-23-05654-f001:**
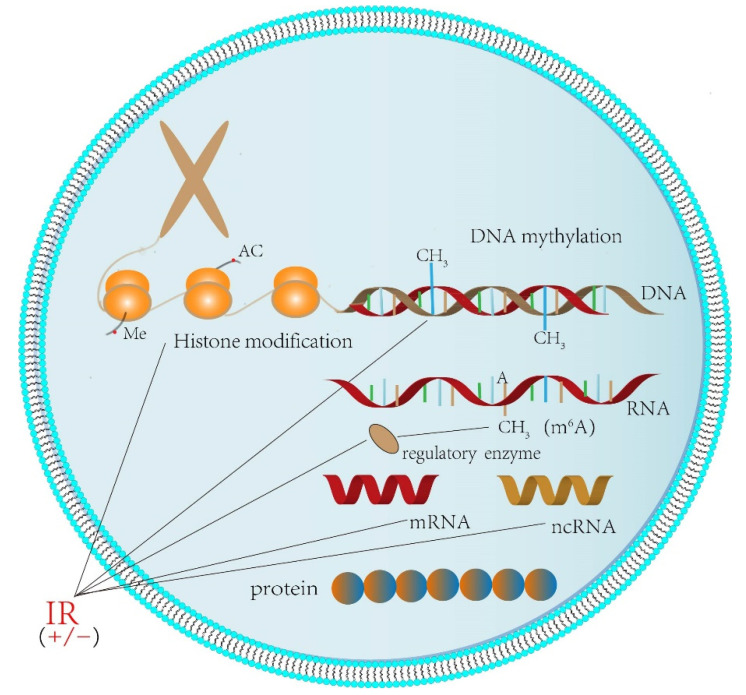
Chromosomes are genetic material in cells and are composed of DNA and proteins, and proteins are mainly histones. DNA can be transcribed and translated into RNA (coding RNA and non-coding RNA) and proteins in cells. Chromatin, histone, DNA and RNA can undergo epigenetic changes such as DNA methylation, histone methylation, histone acetylation and RNA methylation. Studies found that IR can affect cell epigenetics, and it can be applied to DNA and cause DNA methylation levels to be higher or lower. IR also can affect the level of histone methylation, acetylation and tumor cells’ RNA adenosine levels of methylation by acting on enzyme. In addition, it will also affect non-coding RNAs and chromatin remodeling.

**Figure 2 ijms-23-05654-f002:**
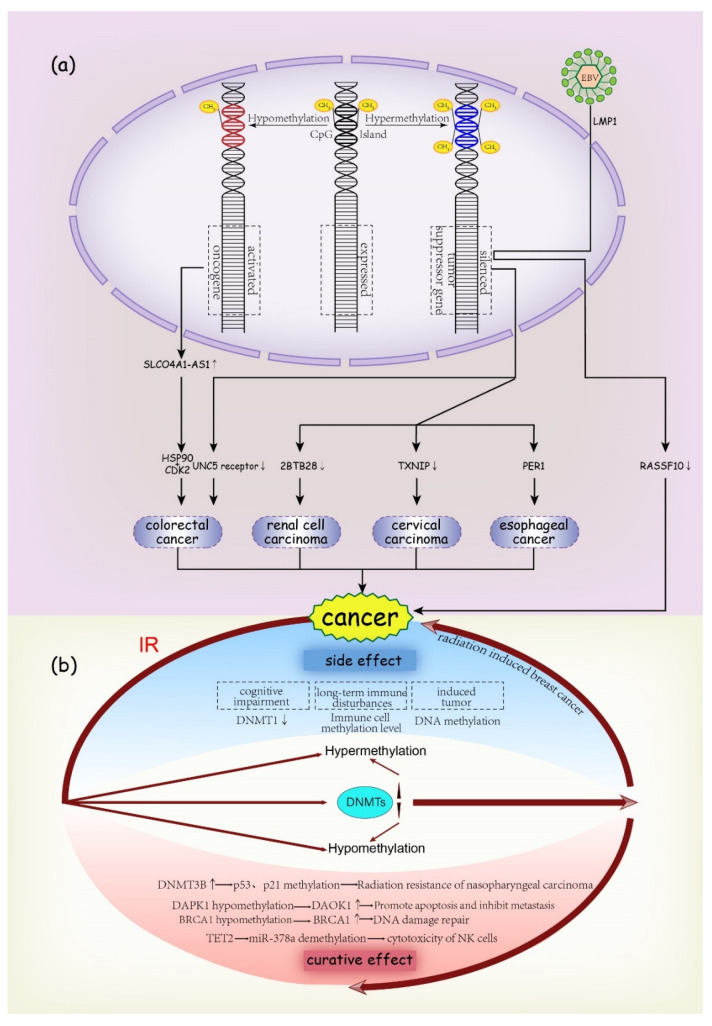
(**a**) DNA methylation changes may be involved in the carcinogenesis and development of cancer, such as hypomethylation leading to the transcriptional activation of oncogenes and hypermethylation leading to transcriptional silencing of tumor suppressor genes. At the same time, viruses may also induce cancer through this mechanism. (**b**) When radiotherapy is used in the treatment of tumors, radiotherapy may also cause different changes in DNA methylation status and DNMT content. These changes may have side effects on the body or have a therapeutic effect. DNMT: DNA methyltransferase. EBV: Epstein-Barr virus.

**Figure 3 ijms-23-05654-f003:**
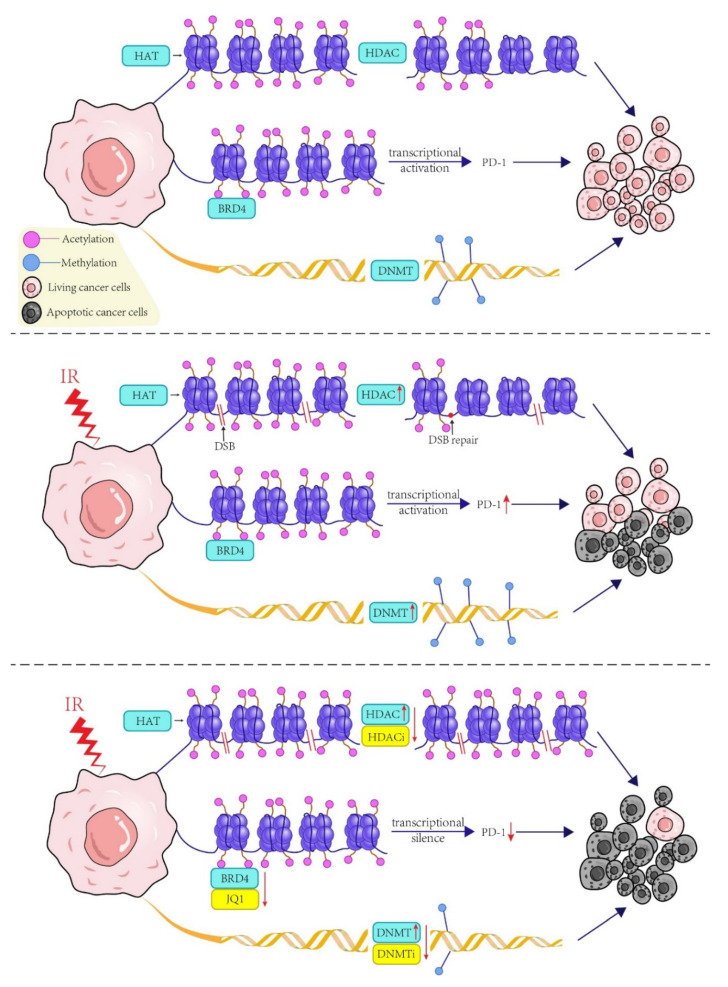
In normal growth and proliferation of tumor cells, histone is regulated by HAT and HDAC acetylation and deacetylation; DNA methylation is regulated by DNMT; BRD4 and CD274 acetylation histone binding to regulate PD-L1 expression, indirectly playing a role in regulating tumor growth. Radiotherapy is an important treatment for tumor, which can promote tumor cell apoptosis by affecting the activities of HDAC, DNMT and PD-L1 to a certain extent. Enhanced activity of HDAC and DNMT promotes histone deacetylation and DNA methylation, leading to accelerated DSB repair. At the same time, radiotherapy can induce the up-regulation of PD-L1 and lead to the escape of tumor cells. In combination with radiotherapy and epigenetic modifiers, HDACi combined with HDAC decreased the activity of HDAC and inhibited DSB repair. DNMT inhibitors promote cell damage by binding DNMT; as a BRD4 inhibitor, JQ1 plays a role in reducing the expression of PDL1 induced by radiotherapy. All of them can enhance the apoptosis effect and play the role of radiotherapy sensitization. DNMTi: DNMT inhibitors.

**Table 1 ijms-23-05654-t001:** The effect of different types of miRNA on cancer and the effect of radiotherapy on miRNA.

MiRNA	Cancer	Changes and Effects in Cancer before Radiotherapy	Radiation Therapy Affects Its Expression	References
MiR-21	Bladder cancer, breast cancer	Up-regulation, promoting cancer progression, migration and invasion	Increase or decrease	[79,92,93]
MiR-155	Bladder cancer, breast cancer and nasopharyngeal cancer	Up-regulation, promote cancer proliferation and poor prognosis	Increase	[79,80,94,95]
MiR-224	Non-small-cell lung cancer, colorectal cancer and bladder cancer	Up-regulation, promoted cancer proliferation and predicted disease course markers	-	[79,96,97]
MiR-196a	Prostate cancer, gastric cancer	Up-regulation, with increased radiosensitivity	Decrease	[98,99]
Let-7	Breast, lung, colorectal cancer	Up-regulation, with increased radiation sensitivity	Decrease	[93,100,101]
MiR-142-3p	Breast cancer and Colon cancer	Up-regulation, to stimulate the apoptosis-related genes	Increase	[102,103,104]
MiR-142-5p	Gastric cancer, esophageal cancer	Down-regulation, promotes macrophage apoptosis and is closely related to cancer development	Increase	[103,105,106]

**Table 2 ijms-23-05654-t002:** Epigenetic modifiers currently used in cancer.

Sensitizer	Type	Target Spot	Cancer	Reference
5-Aza	DNMT inhibitor	DNMT	Ovarian cancer, sarcoma	[149,152]
5-Aza	DNMT inhibitors	P62/SQSTM1	Head and neck cancer	[153]
SGI-110	DNMT inhibitor	DNMT	Sarcoma	[152]
MS-275	HDAC inhibitors	P62/SQSTM1	Head and neck cancer	[153]
Entinostat	HDAC inhibitors	FILP	Non-small cell carcinoma	[144]
5-Aza-Cdr	DNMT inhibitors	RUNX3/TLR9	Carcinoma of the lungs	[154]
C-7280948	PRTM1 inhibitors	PKP2	Carcinoma of the lungs	[155]
JQ1	BRD4 inhibitors	PD-L1	Non-small cell carcinoma	[156]
JQ1	BRD4 inhibitors	RAD51AP1	Cervical carcinoma	[157]
RRX-001	G-6-PD inhibitors	G -6-PD	U87 tumor	[158]

The full name represented by the abbreviation in the Table 2: 5-Aza: 5-Aza-2’-deoxycytidine (Decitabine); SGI-110: Guadecitabine; MS-275: Entinostat (MS-275); 5-Aza-Cdr: 5-Aza-2’-deoxycytidine (Decitabine); JQ-1: JQ-1 (carboxylic acid); G-6-PD: Glucose-6-phosphate dehydrogenase; FILP: FLICE inhibitory protein; PD-L1: Programmed cell death-Ligand 1.

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
