# Peer review of "The Advances in Epigenetics for Cancer Radiotherapy"

_ijms, 2022, doi:10.3390/ijms23105654_

Round 1

Reviewer 1 Report

In this review, the authors have reviewed the mechanism of interaction between epigenetic modification and cancer radiotherapy. The authors concluded that the mechanism of cancer induced by exposure to drugs, viruses, organic chemicals, and other risk factors may be related to various epigenetic pathways.  In the current reviewer’s opinion, the manuscript may be considered for a minor revision.

Comments to the Authors

  1. In introductions, authors may give more details about epigenetic drugs which work well with radiation and chemotherapy treatment for cancer.
  2. Authors may give more details of chemotherapy treatment with suitable references in section 4.
  3. Authors may modify the figures for a better understanding of the readers especially increase the font size.
  4. Authors may include figures in section 4 i.e, the Application of epigenetic modifiers in the sensitization of radiotherapy.

Reviewer 2 Report

Manuscript title:        The Advances in Epigenetics for Cancer Radiotherapy

Journal:                   International Journal of Molecular Sciences

Manuscript ID :  ijms-1713466

Dear Editor: 

Thank you to send me this manuscript. I wrote my evlution below:

(1) Present original findings, conclusions or analysis that has not been published previously by the authors or others : Yes

  (2) Written clearly: No

 (3) Have a high impact in its subfield: Yes

This manuscripte (Investigation the Effect of Series Resistance on the Electrical Parameters of Solar Cell using Multisim Software). The work presented here is interesting and scalable. It is a very relevant subject of study. I would recommend publication of this manuscript in this journal after the authors rigorously addressed some comments listed below.

  • The abstract section does not have all important information. I prefer to rewrite this section by fiiling the spesefice and importante informations. So, it has more confuse for me.
  • The introduction section is very good information, the authors do the best for this section.
  •  There are several passages of the manuscript which are incorrect.

Finally, if the authors do all these comments, I recommend to publish this manuscript in this journal.

        Best Regards

Assist. Pro. Dr.Mohammed H. Mohammed
